# PROPICKER: PROMPTABLE SEGMENTATION FOR PARTICLE PICKING IN CRYOGENIC ELECTRON TOMOGRAPHY

## ABSTRACT

Cryogenic electron tomography (cryo-ET) can produce detailed 3D images called tomograms of cellular environments. An essential step of cryo-ET reconstruction and analysis is to find all instances of a protein in tomograms, a task known as *particle picking*. Due to the low signal-to-noise ratio, artifacts, and vast diversity in proteins, particle picking is a challenging 3D object detection problem. Existing approaches are either slow or limited to picking a few particles of interest, which requires large annotated and difficult to obtain training datasets. In this work, we propose ProPicker, a fast and universal particle picker that can detect particles beyond those in the training set. Our promptable design allows for selectively detecting a specific protein in the volume based on an input prompt. Our experiments demonstrate that through a favorable trade-off between performance and speed, ProPicker can achieve performance close to or on par with state-of-the-art universal pickers, while being up to an order of magnitude faster. Moreover, ProPicker can be efficiently adapted to new proteins through fine-tuning with a few annotated samples.

## 1 INTRODUCTION

Cryo-electron tomography (cryo-ET) is very popular due to its unique capabilities of imaging biological macromolecules in their native environments (Turk & Baumeister, 2020; Hylton & Swulius, 2021). An ambitious goal of cryo-ET is to obtain an 'atlas' of the cell with all of its constituent macromolecules mapped in their native environment. This would revolutionize our understanding of essential protein interactions and has the potential to provide breakthroughs in modern medicine spanning cell biology to drug discovery (Bodakuntla et al., 2023).

In this paper, we focus on particle picking, which is the task of finding all instances of a particle of interest in 3D volumes, called tomograms, obtained with cryo-ET. Particle picking is an essential step in cryo-ET reconstruction and analysis.

Particle picking is a challenging 3D object detection problem, for various reasons. First, due to the fundamental limitations of data acquisition in cryo-ET, tomograms have a very low signal-to-noise ratio and exhibit strong artifacts. Moreover, tomograms are often large ($200 \times 1000 \times 1000$ voxels and up), and cryo-ET datasets can consist of more than a hundred tomograms, which makes their analysis computationally demanding (Genthe et al., 2023; Zeng et al., 2023). Finally, due to the significant diversity in protein types within the cell, there is a vast array of unique object classes to be detected, many of which only differ subtly, rendering differentiation challenging. For instance, the human body alone is estimated to contain more than 20,000 unique proteins (Li & Buck, 2021).

Given these challenges, a particle picking method should be accurate, fast, and universal, i.e., should be able to pick any particle of interest. Existing methods for particle picking are either slow or not universal, that is they are limited to picking a few (and fixed) particles of interest.

In this paper, we propose ProPicker, a **Pro**mptable particle **Picker** that can rapidly detect any type of protein selected by a versatile prompting mechanism. ProPicker is inspired by the Segment Anything Model (SAM) (Ravi et al., 2024; Kirillov et al., 2023) and CLIPSeg Lüddecke & Ecker (2022). For fast particle picking, ProPicker leverages an efficient 3D segmentation network to segment particles

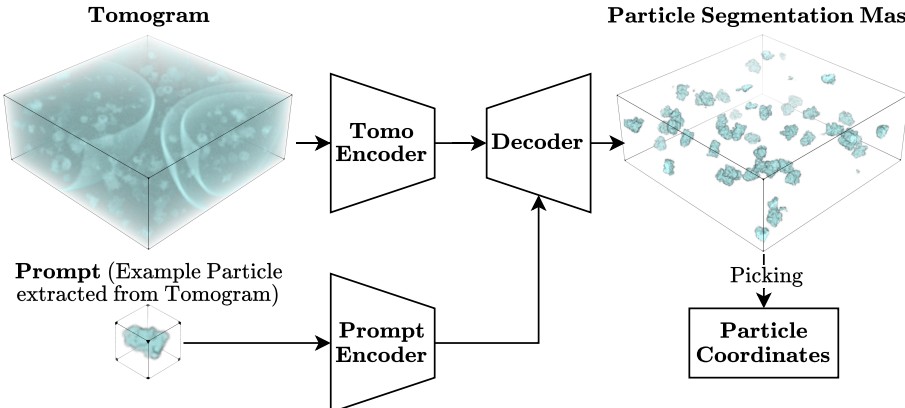

Figure 1: Overview of ProPicker: We extract a generalized representation of the particle to be picked in the tomogram by a prompt encoder. Conditioned on the prompt features, we segment the tomogram for the desired particle. Finally, we leverage the output segmentation map to find the particle coordinates, either by clustering or template-based approaches. The 3D objects shown in this figure have been rendered using data from the SHREC 2021 dataset (Gubins et al., 2020).

of interest in tomograms and accurately locate their positions. To make ProPicker universal, we propose a promptable segmentation architecture that uses a conditioning mechanism to control the type of particle to be segmented by the network. The prompt provides a generalized representation of the particle one wishes to pick and is not restricted to those encountered during training.

As large and diverse datasets of tomograms with annotated particles for training are not yet available (efforts to collect them are well underway (Ishemgulova et al., 2023)), our paper primarily relies on synthetic data for training and evaluation. We also demonstrate the applicability of ProPicker to real-world tomograms. Our main contributions are as follows:

1. We propose ProPicker, a novel particle picker that offers a favorable trade-off between picking performance and speed. ProPicker can accurately pick a variety of different proteins while simultaneously being up to $10\times$ faster at single particle picking than the state-of-the-art.

2. ProPicker is universal in that it can pick new particles that the model has not encountered during training. For cases where ProPicker's out-of-the-box picking performance on novel particles is not satisfactory, we demonstrate that fine-tuning on little data can increase performance.

3. We provide experiments which indicate that ProPicker can generalize to unseen real-world tomograms, even though we train the model exclusively on synthetic data.

Overall, our findings demonstrate that ProPicker is a fast, high-performance and universal particle picker that is based on the contemporary foundation model paradigm. As such, we expect major further improvements in both performance and generalization in the near future when larger and more diverse datasets for cryo-ET become available.

## 2 BACKGROUND & RELATED WORK

Deep learning methods have already revolutionized particle picking from 2D micrographs in the context of single particle cryo-EM (Wang et al., 2016; Bepler et al., 2019; Wagner et al., 2019). They are also on the rise for picking particles in 3D tomograms produced with cryo-ET. However, especially in cryo-ET, traditional methods still play an important role.

**Particle Picking with Template Matching.** Template Matching (TM) is the most widely used method for particle picking in cryo-ET (Bohm et al., 2000; Cruz-León et al., 2024). TM-based methods compare a template of the particle to be picked to candidate sub-tomograms extracted from

a 3D sliding window. This approach is universal, as it can pick any particle as long as a template is available. However, TM is computationally demanding (up to several hours per tomogram), as the stride of the sliding window needs to be small for accurate picking (Genthe et al., 2023; Maurer et al., 2024).

Building upon classical template-based approaches, the recent TomoTwin method (Rice et al., 2023) utilizes a learned convolutional encoder to map both template and sub-tomogram into a structured latent space, where similarity is evaluated. The training of the encoder follows a deep metric learning approach such that latent representations of particles of the same class have high cosine similarity, while those of different classes have low cosine similarity. TomoTwin is state-of-the-art among universal particle pickers, can be faster than classical TM, and is more convenient to use.

**Particle Picking with Deep Learning-Based Object Detection.** Particle pickers using deep learning-based object detection often outperform TM in terms of performance and picking speed (Gubins et al., 2020; Genthe et al., 2023). Many such pickers, including our ProPicker method, use a convolutional network to segment particles of interest belonging to one or more classes and produce candidate particle locations by clustering the predicted segmentation masks. Examples for such methods include DeepFinder (Moebel et al., 2021), DeePiCt (De Teresa-Trueba et al., 2023), and DeepETPicker (Liu et al., 2024).

Deep learning-based object detection approaches for particle picking are typically significantly faster than TM-like methods (Gubins et al., 2020), but current variants are not universal: they are trained on datasets containing a fixed set of particles of interest and trained models are limited to picking particles seen during training.

As a first step towards a universal segmentation-based picker, Zhao et al. (2024) have recently proposed CryoSAM. CryoSAM takes user-specified center coordinates of target proteins as input. Leveraging DINOv2 (Oquab et al., 2023), it extracts features around these points and generates new prompts via feature matching with other parts of the tomogram. The general 2D segmentation model SAM (Kirillov et al., 2023) then segments particles iteratively, slice by slice, using the newly generated prompts.

**Relation of ProPicker to Existing Particle Pickers.** ProPicker shares the setup of picking particles based on a single observation with TomoTwin and CryoSAM. In contrast to TomoTwin, ProPicker does not rely on slow template matching, but is based on fast 3D segmentation. CryoSAM is segmentation-based but, the techniques used differ significantly from ProPicker. CryoSAM does not involve any training or fine-tuning on cryo-ET specific data but relies on 2D models trained in the general domain on mostly natural images. ProPicker on the other hand uses 3D segmentation and has been trained on domain-specific cryo-ET data. We compare the performance and speed of TomoTwin, CryoSAM and ProPicker in Section 4.

## 3 ProPicker: Promptable Segmentation for Particle Picking

After stating the problem we consider in most parts of this work, we describe ProPicker (Figure 1).

**Problem Statement** We are given a set of tomograms $\{\boldsymbol{x}_1, \boldsymbol{x}_2, ...\}$, represented as 3D voxel arrays, i.e., $\boldsymbol{x}_i \in \mathbb{R}^{n \times n \times n}$. Assume that the tomograms contain multiple instances of a type of particle in which we are interested, and whose size is known. The individual instances can appear at arbitrary orientations. We consider the task of finding the set $\mathcal{C}^* = \{\boldsymbol{c}_1^*, \boldsymbol{c}_2^*, ...\}$ containing the centers $\boldsymbol{c}_i^* \in \mathbb{R}^3$, $i = 1, 2, ...$ of all instances of the particle of interest in all of the tomograms.

### 3.1 ProPicker architecture

ProPicker takes a tomogram $\boldsymbol{x}$, and a prompt, a 3D voxel array $\boldsymbol{p} \in \mathbb{R}^{m \times m \times m}$ representing the particle of interest as input. The output is a set $\mathcal{C}$ containing predicted particle centers.

In this work, a prompt $\boldsymbol{p}$ is a small sub-tomogram, i.e., a part of a larger tomogram, of shape $37 \times 37 \times 37$ containing one instance of the particle of interest. The prompt sub-tomogram is extracted from one of the tomograms to which we wish to apply particle picking. However, other

representations that uniquely identify particles, such as amino-acid sequences or atomic coordinates could also be used as prompts.

ProPicker consists of a prompt encoder and a segmentation model, which we detail in the following.

**The prompt encoder.** The prompt encoder $\varepsilon_{\boldsymbol{p}} : \mathbb{R}^{m \times m \times m} \to \mathbb{R}^d$ extracts a salient feature vector $\boldsymbol{z_p} \in \mathbb{R}^d$ that encodes information required for efficiently detecting the particle in tomograms. We focus on prompts in voxel-space and use the TomoTwin encoder (Rice et al., 2023) as the prompt encoder due to its robustness and good performance in template-based particle classification (see Section 2). Its input is a sub-tomogram that includes the particle of interest, and it outputs a concise ($d = 32$) representation $\boldsymbol{z_p}$ of the particle that we use to condition the segmentation model.

**The segmentation model.** Given an input volume $\boldsymbol{x} \in \mathbb{R}^{n \times n \times n}$ and prompt $\boldsymbol{p}$, our promptable segmentation model $\mathcal{S} : \mathbb{R}^{n \times n \times n} \times \mathbb{R}^d \to \mathbb{R}^{n \times n \times n}$ can be conditioned on the input prompt that steers the output map $\boldsymbol{y} \in \{0, 1\}^{n \times n \times n}$ to the desired particle class, that is $\boldsymbol{y} = \mathcal{S}(\boldsymbol{x}; \boldsymbol{z_p})$, where $\boldsymbol{z_p} = \varepsilon(\boldsymbol{p})$. The model output $\boldsymbol{y}$ is the voxel-wise prediction of the model with respect to the absence/presence of the particle described in the input prompt. promWe use a well-established convolutional 3D U-Net (Ronneberger et al., 2015), which is an encoder-decoder architecture (see Figure 1), as our segmentation model. The U-Net's encoder consists of 5 spatial downsampling layers, and the corresponding decoder has 5 spatial upsampling layers. In total, the U-Net has 124 million trainable parameters. More details on the segmentation model and how we condition it on the encoded prompt can be found in Appendix B.

## 3.2 Particle Picking Pipeline with ProPicker

Next, we describe the steps of picking a single particle type in a tomogram using ProPicker.

**1. Prompt extraction.** First, we obtain a representation of the particle of interest in the tomogram. We focus on voxel-space prompting, and therefore manually extract a sub-tomogram that includes an instance of the particle of interest to be used as a prompt.

**2. Particle segmentation.** Next, we embed the extracted prompt and segment the tomogram using ProPicker. As tomograms are typically very large, we segment the volume using a strided 3D sliding window approach. Specifically, we slide a moderately sized window across the tomogram to extract sub-tomograms, and segment each sub-tomogram individually. We obtain a full-sized segmentation mask for the tomogram by combining the sub-tomogram level masks, averaging overlapping regions. An important hyperparameter of particle segmentation is the stride $s$ of the sliding window, which controls the overlap between segmented sub-volumes. Template-based pickers rely on high overlap (low $s$), as these techniques need the window to be centered around the target particle. Our segmentation-based approach is less sensitive to the amount of overlap, and thus performance can be maintained with larger strides. Due to the 3D nature of the problem, a factor $N$ reduction in stride results in $N^3$ times decrease in compute. Note that it is also possible to use strategies other than the sliding window to process the tomogram, which may potentially yield higher speed and/or improved segmentation masks (Rumberger et al., 2021).

**3. Finding particle center coordinates.** We propose two strategies to map the segmentation output $\boldsymbol{y}$ to particle center coordinates:

- **Clustering-based picking (ProPicker-C):** We detect clusters in the segmentation map by finding connected components. The centroid of each cluster is a predicted particle center. The precision of this approach can be improved by leveraging prior information about the target particle size by excluding clusters that are too small or too large.

- **TM-based picking (ProPicker-TM):** We apply a TM-based picker to the input tomogram over regions where our segmentation mask predicts the presence of a particle.

Our default variant, ProPicker-C, is fast as it directly predicts the particle centers from the segmentation map using lightweight connected component finding.

The performance of ProPicker-TM depends on the TM procedure which produces the actual picks. Therefore, ProPicker-TM is particularly useful when the goal is to speed-up an existing TM pipeline in case the latter is known to have good performance. As the segmentation masks produced with ProPicker are typically very sparse in the volume (in our experiments, up to 98% of the tomograms were masked out), ProPicker-TM can yield significant speedups for TM.

## 4 EXPERIMENTS

In this Section, we demonstrate that ProPicker is effective at picking particles. We consider two different setups, divided into Section 4.1 and Section 4.2.

In Section 4.1, we consider picking all copies of a particle based on a single observation (prompt) of the particle. As baselines, we consider TomoTwin (Rice et al., 2022) and CryoSAM (Zhao et al., 2024). However, we find that TomoTwin performs better than CryoSAM and that CryoSAM is unable to handle diverse, crowded tomograms, and is less robust to noise (see Section 4.1.2). Therefore the state-of-the-art TomoTwin is our main baseline. We do not compare to non-universal methods or methods that require *any* data from the target tomograms for training, as these do not match the setup. We compare to such methods in Section 4.2.

In Section 4.2, we discuss fine-tuning ProPicker as a means to improve performance and to adapt to new particles. This setup is similar to that of non-universal deep-learning-based pickers, out of which we choose the popular and powerful DeepFinder method (Moebel et al., 2021) as baseline.

Before we discuss our experiments, we describe the training of ProPicker and our evaluation metrics.

**Training Dataset of ProPicker.** We train ProPicker on realistically simulated tomograms from Rice et al. (2023) and Gubins et al. (2020), which have also been used to train TomoTwin. Our training set contains the majority of TomoTwin's training data, and consists of 78 tomograms containing a total of 113 unique protein types, as well as gold fiducial markers, vesicles and filaments. Each tomogram contains around 1500 protein instances, each belonging to a set of $10 - 13$ unique protein types. We train on sub-tomograms of size $64 \times 64 \times 64$ extracted from all tomograms with a 3D sliding window with stride 32.

**Training ProPicker.** To reduce computational cost, we keep the prompt encoder frozen during training (see Section 3.1). We train the segmentation model with the Adam optimizer (Kingma & Ba, 2015) with fixed learning rate 0.01. For each gradient step, we first randomly sample a batch of 8 sub-tomograms. For each sub-tomogram, we also randomly sample 10 prompts, one for each of the 10 unique proteins the sub-tomogram may contain (see "Training Dataset"). We pass each sub-tomogram and its corresponding prompts through the conditional segmentation model. This yields a total of $80 = 8 \cdot 10$ predicted segmentation masks. Finally, we compute the average voxel-wise binary cross entropy between the model outputs and the 80 single-class target masks as loss.

**Evaluation Metrics.** We measure picking performance with F1 score, and report best-case performance with thresholds optimized on test data for all methods following common practice (Rice et al., 2023), unless stated otherwise.

### 4.1 PICKING PARTICLES BASED ON A SINGLE PROMPT

ProPicker targets fast and universal picking. We first focus on picking speed and compare to TomoTwin which achieves state-of-the-art performance for the dataset we consider.

### 4.1.1 PICKING SPEED

Both TomoTwin and ProPicker process the tomogram in a 3D sliding window fashion. Therefore, inference time is cubically related to the window stride. However, large strides (small overlap) often results in low detection performance. Here, we explore this trade-off. To quantify speed, we report the throughput in tomograms per hour on a single NVIDIA L40 GPU for picking a single particle of interest in a tomogram of size $200 \times 512 \times 512$.

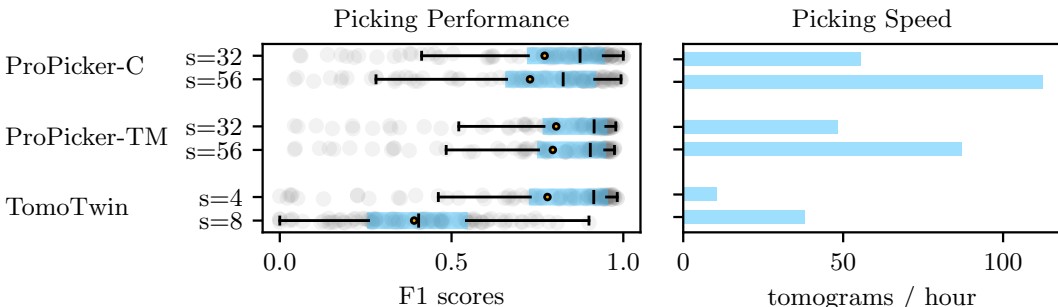

Figure 2: Picking performance (best-case F1 scores) and speed for our method with ProPicker-C, ProPicker-TM and TomoTwin (TT) for 100 unique particles seen during training. Vertical markers are medians, circles are means.

| Method / PDB | 1avo | 1e9r | 1fpy | 1fzg | 1jz8 | 1oao | 2df7 | Mean | Median |
|---|---|---|---|---|---|---|---|---|---|
| CryoSAM | 0.21 | 0.30 | 0.43 | 0.22 | 0.50 | 0.45 | 0.70 | 0.40 | 0.43 |
| TomoTwin | 0.59 | 0.72 | **0.86** | 0.48 | 0.73 | **0.70** | 0.92 | 0.71 | 0.72 |
| ProPicker-C | **0.66** | **0.82** | 0.76 | **0.55** | **0.79** | **0.70** | **0.98** | **0.75** | **0.76** |

Table 1: Best-case F1 picking scores for CryoSAM, TomoTwin ($s = 2$) and ProPicker-C ($s = 32$) on 8 particles that were not seen during training.

As the speed at which a particle can be reliably picked depends on, e.g., the particle's size (especially for TM methods like TomoTwin), we measure picking speed on a set of 10 tomograms which contain 100 particles in total. Both TomoTwin ProPicker and have seen all of these particles during training, but within in different tomograms, i.e., in different contexts. We evaluate performance on unseen particles in Section 4.1.2.

As can be seen in Figure 2, ProPicker-C with $s = 32$ can pick most particles as well as TomoTwin for $s = 4$ (TomoTwin's default $s = 2$ gives slightly better performance but is almost $8\times$ slower), while being more than $5\times$ faster. Increasing ProPicker's stride to $s = 56$ doubles the speed while resulting in a moderate loss of performance. Note that TomoTwin cannot be significantly accelerated by increasing the stride, as even $s = 8$ leads to a large drop in performance (Fig. 2, bottom left).

ProPicker-TM (with TomoTwin TM) with $s = 56$ increases throughput by a factor of 10 at the cost of a small decrease in performance over TomoTwin with $s = 4$. This shows that ProPicker-TM can maintain the accuracy of the high-performing TM-based picker TomoTwin, while providing large speedups by reducing TomoTwin's search space to the segmentation mask predicted by ProPicker.

ProPicker's speed advantage over TomoTwin, and TM in general, is due to the efficient segmentation with the 3D U-Net which. Other picking methods that use convolutional models for segmentation like DeepFinder (Moebel et al., 2021), DeePiCt (De Teresa-Trueba et al., 2023) or DeepETPicker (Liu et al., 2024) offer favorable speed-performance tradeoffs similar to ProPicker, but are strictly limited to picking particles seen during training.

### 4.1.2 GENERALIZATION TO UNSEEN PARTICLES

Next, we study the universality of our picker, i.e., its capability to generalize to unseen particles. We test the generalization capability of ProPicker on a re-generated version of the generalization tomogram from Rice et al. (2023). This tomogram contains 8 unique particles that are not part of our training set, we generate it with the same simulator as most of ProPicker's and TomoTwin's training dataset. The tomogram is therefore well suited for studying the generalization of ProPicker to unseen particles in an environment similar to that seen during training.

Our setup in this section is also well-suited to compare methods for universal picking. Therefore, we consider TomoTwin and CryoSAM as baselines. We applied TomoTwin and ProPicker directly to the tomogram using the same prompts. We find that CryoSAM is unable to pick particles in the noisy tomogram. Therefore, we applied CryoSAM to the clean ground truth tomogram that underlies the simulation of the noisy tomogram to probe the best-case performance.

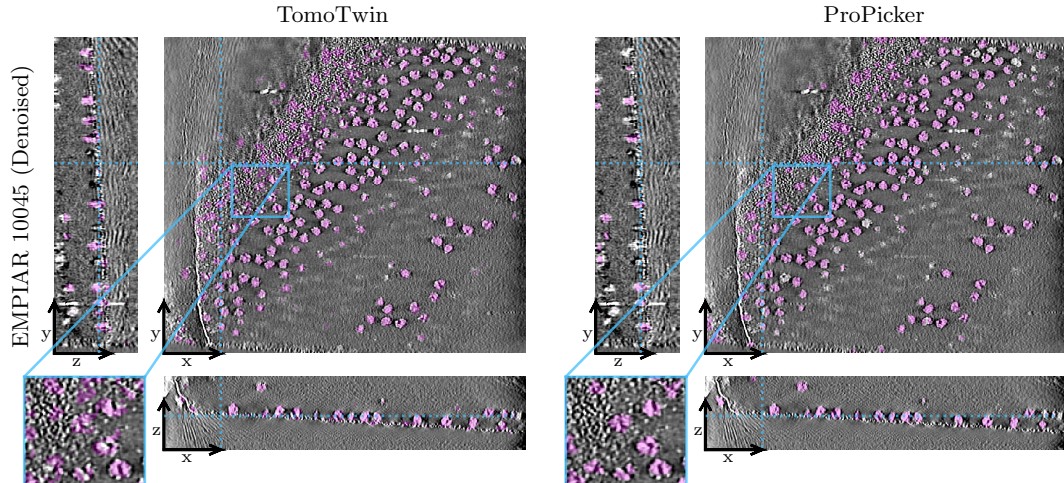

Figure 3: Slices through a single real-world tomogram from EMPIAR 10045. TomoTwin's thresholded heatmap and our segmenation map are shown in transparent pink. For clarity, we denoise the tomogram with CryoCARE (Buchholz et al., 2019); the CryoCARE-denoised tomograms are not used for picking. The images are high resolution, and we recommend readers to zoom in to judge the picking performance of the methods.

As shown in Table 1, for these 8 particles, ProPicker-C ($s = 32$) performs on par with TomoTwin ($s = 2$). This demonstrates that ProPicker-C pre-trained on a set of particles can efficiently generalize to new proteins by providing input prompts in a format compatible with the training setup.

CryoSAM achieves lower F1 scores than TomoTwin and ProPicker for all particles. Upon inspection, we find that CryoSAM's segmentation masks confuse particles. We conjecture that both DINOv2 (Oquab et al., 2023) and SAM (Kirillov et al., 2023), which are both used in CryoSAM without any fine-tuning on cryo-ET data, fail to discriminate between the particles even in the absence of noise, further highlighting the need for domain-specific training for accurate particle picking.

### 4.1.3 GENERALIZATION TO REAL-WORLD TOMOGRAMS

To further test the generalizability and universality of ProPicker, we perform case studies on real-world tomograms. Here, we only compare to TomoTwin which we find to be the stronger baseline. Results for CryoSAM are in Appendix D.

First, we consider EMPIAR 10045, consisting of tomograms of purified 80S ribosomes, i.e. ribosomes embedded in ice without cellular context or other proteins (Bharat & Scheres, 2016). We also consider EMPIAR 10988, containing tomograms with ribosomes within *S. pombe* cells (De Teresa-Trueba et al., 2023). For both datasets, we picked ribosomes using ProPicker (with $s = 32$) and TomoTwin (with $s = 4$).

Figure 3 and Figure 4 show segmentation masks without clustering or TM-based picking produced with ProPicker alongside segmentation masks produced with TomoTwin. The raw output of both TomoTwin and ProPicker are 3D volumes with voxel values between 0 and 1. For ProPicker this is the output of the segmentation U-Net (after sigmoid activation). For TomoTwin, the raw output is a 3D heatmap which, at each voxel, contains the cosine similarities between the embedded candidate sub-tomogram centered at this voxel and the embedded reference. We threshold the output volumes to obtain the binary-valued segmentation masks in Fig. 3 and 4.

We find that, for both real-world tomograms, ProPicker performs better when applied to slightly denoised tomograms. In both cases, we applied very mild denoising with a Gaussian filter whose kernel has a standard deviation of 0.6 for EMPIAR 10045 and a standard deviation of 0.5 for EMPIAR 10988. Note that for the synthetic tomograms, denoising was not necessary, and we believe that the benefit of Gaussian denoising is largely caused by changing the distribution of the noise on the real-world data tomograms such that it is closer to the synthetic noise.

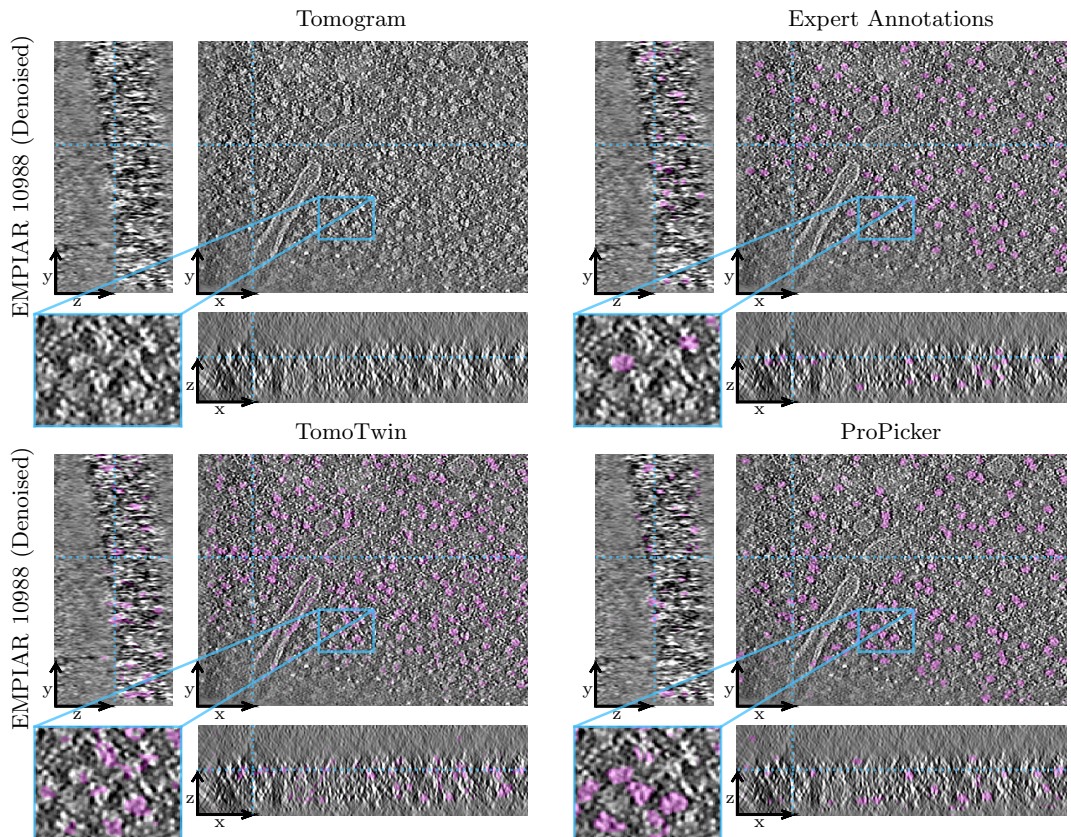

Figure 4: Slices through a single real-world tomogram from EMPIAR 10988. TomoTwin's thresholded heatmap and our segmentation map are shown in transparent pink. For clarity, we denoise the tomogram with Topaz (Bepler et al., 2020); the Topaz-denoised tomograms are not used for picking. The images are high resolution, and we recommend readers to zoom in to judge the picking performance of the methods.

In the EMPIAR 10045 tomogram (Figure 3), both TomoTwin and ProPicker pick a substantial amount of the ribosomes. Note that although it seems that both methods produce false positives in the noisy region shown in the x-y-plane (large panels), an inspection of the x-z-plane reveals that the noisy region actually contains ribosomes which are correctly identified.

The (defocus-only) tomogram from EMPIAR 10988 shown in Figure 4 is more challenging for particle picking due to the crowded cellular environment. Still, both TomoTwin and ProPicker detect many ribosomes. When comparing to the expert annotations (top right panel) by De Teresa-Trueba et al. (2023), we find that most picks of TomoTwin and ProPicker are true positives. For a more quantitative evaluation, we compute best-case picking F1 scores with respect to the expert annotations for both TomoTwin and ProPicker. TomoTwin achieves a best-case F1 of 0.60, ProPicker-C achieves 0.61 on the denoised tomogram and 0.55 on the raw, un-denoised tomogram.

In in Appendix F, present results which indicate that ProPicker can pick particles other than ribosomes as well and is able to distinguish particles in a diverse real-world tomgram.

## 4.2 FINE-TUNING PROPICKER

ProPicker is a generalist model trained to pick a wide range of particles. To improve the picking performance on specific particles or tomograms, ProPicker can be fine-tuned. Here, we show that ProPicker's performance on unseen particles can be signficiantly, and data-efficiently improved through fine-tuning. Note that fine-tuning ProPicker is different from the setup of universal picking based on a single prompt we have considered so far in this work.

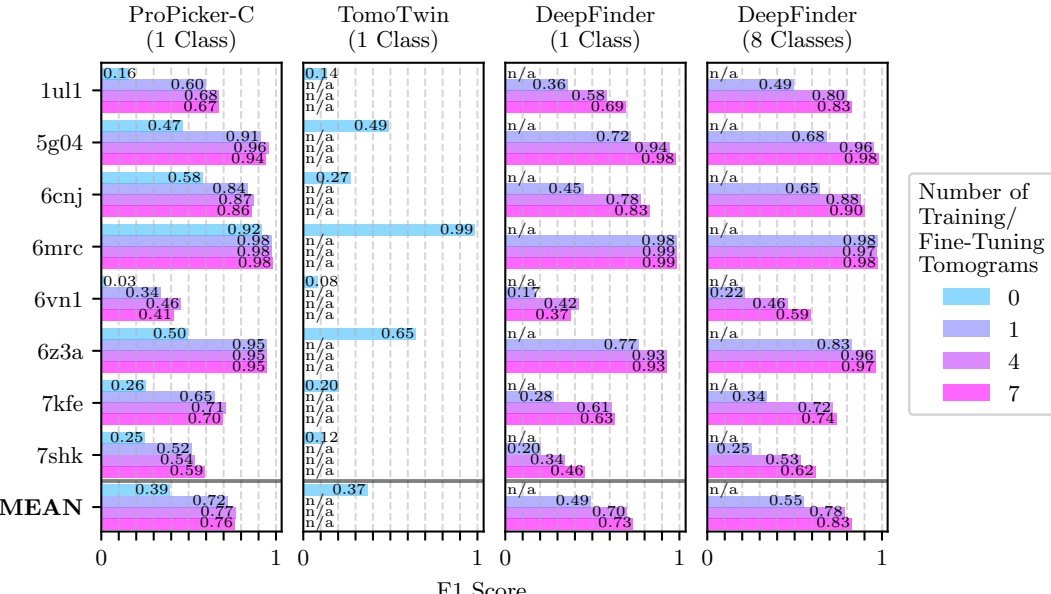

Figure 5: Best-case F1 scores of ProPicker-C and DeepFinder versus amount of fine-tuning data for unseen particles.

**Fine-tuning strategy for ProPicker.** We describe how to fine-tune ProPicker to pick a single particle of interest. This requires (parts of) one or more tomograms with corresponding ground-truth binary segmentation masks of the particle, as well as a manually extracted example of the particle that serves as prompt. During fine-tuninig, we keep this prompt and the prompt encoder fixed, and only fine-tune the segmentation model and the prompt conditioning mechanism.

**Dataset.** We resort to a set of tomograms from TomoTwin's training set each of which contains instances of 8 unique particles. As TomoTwin's training set was specifically desinged to contain particles whose structures are all very different from one another (see (Rice et al., 2023) for details), these 8 particles are hard, unseen examples for ProPicker. We have access to 8 tomograms, out of which we use 7 for fine-tuning and one for testing. All tomograms contain around 150 instances of each particle. For the experiment in this Section, we re-trained ProPicker and its TomoTwin prompt-encoder, and excluded the 8 particles from the both training sets.

**DeepFinder baseline.** We choose DeepFinder (Moebel et al., 2021) as baseline from the class of non-universal state-of-the-art deep learning-based particle pickers. DeepFinder is segmentation-based and uses a 3D U-Net-like convolutional architecture, like many other such pickers, e.g, DeeP-iCt (De Teresa-Trueba et al., 2023) or DeepETPicker (Liu et al., 2024).

We compare to two variants of DeepFinder. For the first variant, "DeepFinder (1 Class)", we train one DeepFinder model for each particle separately. This is the same setup as when we fine-tune ProPicker. For the second variant, "DeepFinder (8 Classes)", we train one DeepFinder model to pick all 8 particles *simultaneously*. Moebel et al. (2021) observed that multi-class training can yield substantial improvements in performance for hard-to-pick particles. Note that, in contrast to the single class setup, the multi-class setup requires annotations for *all 8* particles.

**Results.** Due to the particularly challenging data, we observe rather low picking F1 scores for ProPicker-C on most particles when picking with a single prompt (left panel of Figure 5). Note that the (re-trained) TomoTwin, too, struggles with picking, and achieves a mean F1 score similar to ProPicker-C. As it is not straightforward how to fine-tune TomoTwin on individual particles, we only report the performance of the re-trained TomoTwin for prompt-based picking.

Fine-tuning ProPicker-C significantly boosts picking performance for all particles. The performance of the fine-tuned models saturates quickly as more data becomes available: Fine-tuning on a single

tomogram, which contains around 150 instances of each particle, yields significant improvements for all particles, whereas going from 4 to 7 fine-tuning tomograms makes little to no difference.

If only a few tomograms *or* only annotations for the particle of interest are available, fine-tuning ProPicker-C yields superior performance compared to both variants of DeepFinder. When training/fine-tuning on a single tomogram, the fine-tuned ProPicker-C pickers outperform DeepFinder with 1 class (center panel) and 8 classes (right panel). Even as more data becomes available, ProPicker-C performs as well as or better than the single-class DeepFinder models, but the performance gap is narrowing.

The benefit of ProPicker-C's pre-training is not able to outweigh the advantages of multi-class particle picking if enough training data is available: DeepFinder trained on all 8 classes simultaneously performs on par or better than the 8 fine-tuned ProPicker-C pickers when training/fine-tuning on 4 or 7 tomograms. We again emphasize that the price of the improved performance is having to annotate *all* particles in the tomogram even if one is only interested in a single one.

Further reducing the data requirements for fine-tuning ProPicker by incorporating techniques for data-efficient training (see (Huang et al., 2022)) is a promising direction for future work.

## 5 DISCUSSION & CONCLUSION

In this work, we propose ProPicker, a particle picking method for cryo-ET that leverages a promptable segmentation model for rapid and accurate detection of proteins. The core of ProPicker is an efficient 3D segmentation model capable of selectively detecting particles in tomograms based on a prompt, i.e., a concise representation of the particle of interest.

Depending on the particle to be picked, ProPicker can achieve state-of-the-art picking performance, in some cases even if the particle was not seen during training, while at the same time offering significant speedups compared to the popular state-of-the-art template matching baseline TomoTwin.

Through evaluation on two real-world tomograms, we demonstrated (see Section 4.1.3) that ProPicker can generalize to real-world data. In our experiments, we also encountered real-world tomograms and particles for which ProPicker was unable to produce good results. This is not surprising because, due to a lack of large and diverse annotated real-world datasets, we rely exclusively on synthetic data for training. Such issues related to generalization and robustness are not exclusive to ProPicker (Bandyopadhyay et al., 2022): Huang et al. (2024) reported significant drops in performance when applying deep learning-based particle pickers, among them DeepFinder (Moebel et al., 2021)) and TomoTwin, to tomograms whose characteristics are too different from the training data. Note that a concrete example of such a case can be found in Section 4.2, where we showed that both ProPicker and TomoTwin struggle to pick particles that have not been seen during training and that are very dissimilar from all other particles in the training dataset.

However, the fact that ProPicker is indeed able to perform well on both unseen synthetic particles (Section 4.1.2) and real-world tomograms datasets highlights the great potential of the method. It is widely accepted that training on larger and more diverse datasets improves the robustness of deep learning models (Radford et al., 2021; Fang et al., 2022; Lin & Heckel, 2024). Therefore, a promising direction for future work is to collect large datasets of tomograms with ground truth particle annotations for a variety of particles. Large scale efforts to do so have already been initiated, see for example the CryoET Data Portal (Ermel et al., 2024) and (Ishemgulova et al., 2023). Once such datasets become available, incorporating them into the training sets of universal particle pickers like TomoTwin and ProPicker is likely to increase their robustness and performance.

ProPicker is a step towards a fast foundational particle picking model for cryo-ET. We expect particle picking to profit significantly from adopting the foundation model paradigm where large pre-trained models give good picking results out-of-the-box or are fine-tuned on little data, as demonstrated for ProPicker in Section 4.2.

## REPRODUCIBILITY STATEMENT

We provide code to train and evaluate ProPicker, as well as the weights of the trained model. All datasets used in this paper are publicly available.

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

APPENDIX

## A  SUB-TOMOGRAM AVERAGING

Sub-tomogram averaging is a central application of cryo-ET and one of the main motivations for particle picking. The goal of sub-tomogram averaging is to get high-resolution models of particles like proteins based on data from tomograms. The idea is to align and average sub-tomograms that show the same particle, which suppresses the strong noise and artifacts present throughout the tomograms and can yield high-resolution models of the particle (Wan & Briggs, 2016). For this procedure, particle picking is an essential component and bottleneck both in terms of time demands and accuracy.

## B  ARCHITECTURAL DETAILS

### B.1  PROMPTABLE SEGMENTATION MODEL

We use a well-established convolutional 3D U-Net (Ronneberger et al., 2015) as our segmentation model. The U-Net's encoder matches the architecture of the prompt encoder, and consists of 5 spatial downsampling/upsampling layers with corresponding skip connections. We condition the decoder on the encoded prompt $z_p$, using a feature-wise linear modulation (FiLM) approach (Perez et al., 2018) (see Appendix B.2). Conditioning only the decoder enhances our technique with additional efficiency, as the encoder features can be re-used and decoded through multiple prompts in case one wishes to pick more than one type of particle. As is common in segmentation tasks, we activate the output of the U-Net with a sigmoid function to ensure the voxel values are between 0 and 1. At inference, voxels have to be binarized according to a user-defined threshold for clustering via connected component detection.

### B.2  PROMPT CONDITIONING

We condition each of the decoder's 5 spatial upsampling layers with FiLM (Perez et al., 2018), which we implement as follows: Let $C$ be the number of channels (features) of an intermediate 3D feature map after upsampling. First, we multiply the encoded prompt $z_p \in \mathbb{R}^{32}$ with two (learnable) matrices $A, B \in \mathbb{R}^{C \times 32}$. Finally, we map each channel $k \in \{1, ..., C\}$, with an affine transformation with slope $(Az_p)_k \in \mathbb{R}$ and intercept $(Bz_p)_k \in \mathbb{R}$, which gives the conditioned feature map. We use one separate pair of matrices $(A, B)$ for each of the 5 upsampling layers. We use one FiLM conditioning after each up-sampling layer, so there are 5 pairs of matrices $(A, B)$ in total.

## C  DEPENDENCE OF PICKING PERFORMANCE ON THE PROMPT

The user has to manually extract the prompt from a tomogram at inference, and the tomogram typically contains many instances of the particle of interest. Thus, a natural question is how the performance of ProPicker depends on the concrete choice of the prompt. We investigate this prompt-dependence on two synthetic tomograms both containing 10 unique particles which have been all seen during training. The two tomograms are part of the test set described in Section 4.1.1. As the ground truth location of all particles is known for the synthetic tomograms, we randomly sampled 10 prompts for each particle and evaluated the best-case picking F1 score.

As can be seen in Figure A1, for all but few particles, the picking performance of ProPicker-C does not vary much with the concrete choice of the prompt. For some particles like 5ahu and 6tps (both in Tomogram 1), there are outlier prompts which yield significantly lower picking performance compared to the rest. We therefore recommend to try a few prompts for each particle.

## D  FURTHER RESULTS FOR CRYOSAM

We applied CryoSAM with default hyperparameters to the same tomogram from EMPIAR 10045 as in the experiment discussed in Section 4.1.3. CryoSAM did not produce a good segmentation mask

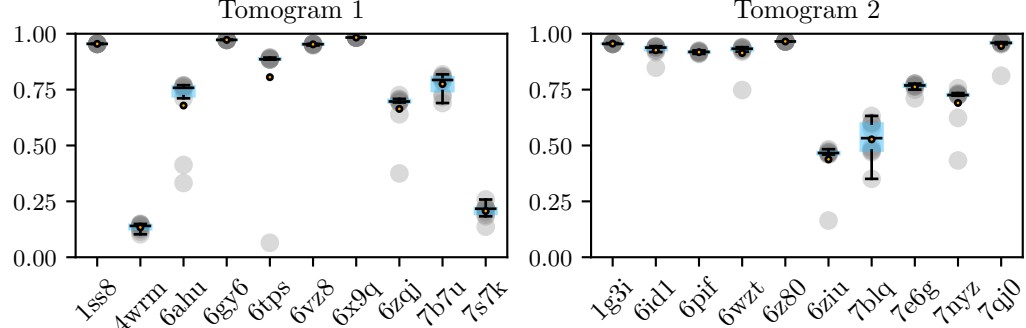

Figure A1: Dependence of the picking F1 score of ProPicker-C on the choice of the prompt for two synthetic tomograms containing particles which were seen during training. The particles are labeled according to their accession code in the protein data bank (PDB).

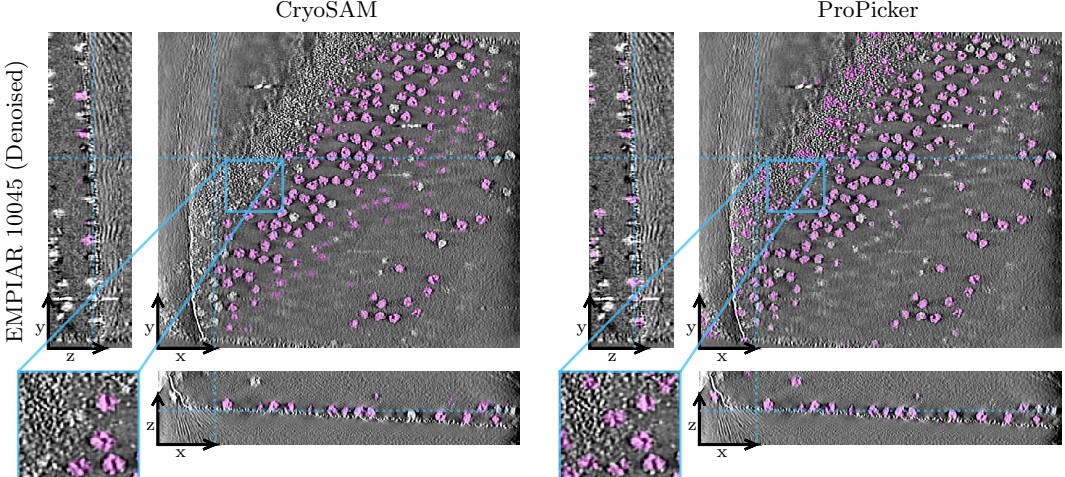

Figure A2: Slices through a single real-world tomogram from EMPIAR 10045. The tomogram underlying the pink segmentation masks has been denoised with CryoCARE. The CryoSAM results have been obtained on the CryoCARE-denoised tomogram, whereas we used Gaussian denoising for ProPicker (see Section 4.1.3).

when applied to the noisy tomogram, so we extensively denoised the tomogram with CryoCARE to produce the results shown in Figure A2. Note that CryoSAM does not pick the particles in those parts of the tomogram that contain structural noise, see, e.g., the diagonal strip in the top right area, or the zoomed-in region.

## E    FINE-TUNING ON SEEN PARTICLES

Finetuning of ProPicker can also improve performance on data seen during training. We consider the synthetic SHREC 2021 dataset (Gubins et al., 2020) which is part of ProPicker's training dataset.

The SHREC 2021 dataset is a popular synthetic benchmark for non-universal particle picking methods which target picking a fixed set of particles of interest. It consists of 10 synthetic tomograms each containing 12 unique proteins. We train one model for each particle (12 models in total) on 8 tomograms. We use the 9th tomogram for early stopping and optimizing the hyperparameters (thresholds, maximal and minimal cluster sizes) of cluster-based picking. The 10th tomogram serves as test tomogram. The test tomogram was part of neither the training dataset nor the fine tuning dataset of ProPicker.

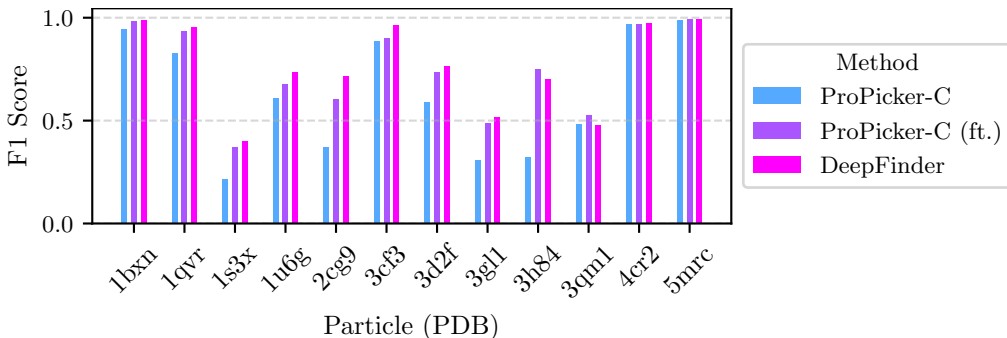

Figure A3: F1 scores for TomoTwin ($s = 2$), DeepFinder and ProPicker-C ($s = 32$) achieved on the SHREC 2021 test tomogram. The F1 scores for DeepFinder were taken from the SHREC 2021 challenge paper by Gubins et al. (2020) (Table 4).

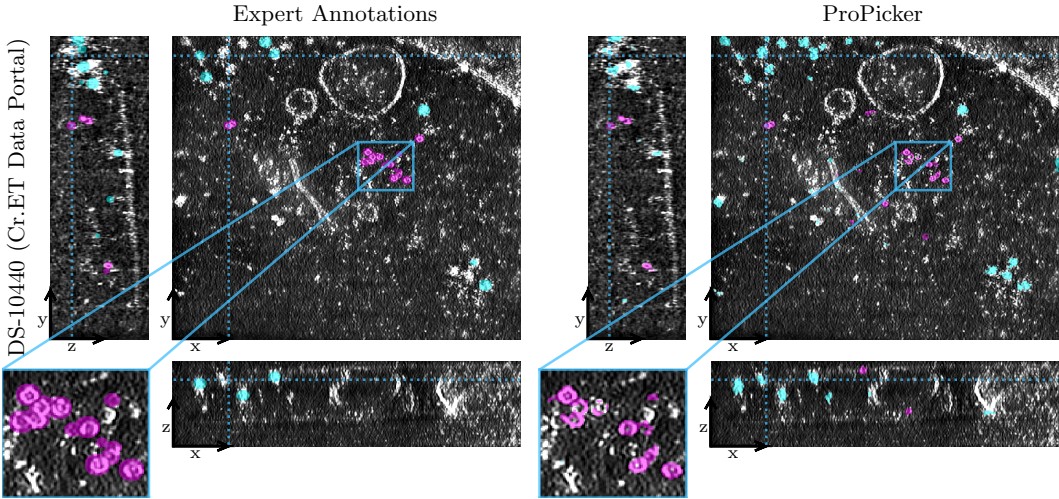

Figure A4: Slices through a single real-world tomogram from the DS-10440 dataset taken from the Cryo ET Data Portal. Ground-truth annotations and ProPicker segmentation masks for apoferritin and cytosolic ribosomes are shown in pink and ice-blue respectively. The tomogram underlying the segmentation masks has been denoised by Peck et al. (2024). The ProPicker masks have been obtained from the raw, un-denoised tomogram (not shown).

Figure A3 shows F1 scores for all particles in the SHREC 2021 dataset. The performance varies widely among particles as some are easy to pick, whereas other particles, especially smaller ones, pose greater challenges (see (Gubins et al., 2020) for an in-depth discussion).

Fine-tuning ProPicker-C on the SHREC 2021 dataset yields large improvements for some particles, especially for the particles for which ProPicker-C without fine-tuning achieves relatively low F1 scores (e.g. 1s3x, 1u6g, 3gl1 and 3h84) can benefit significantly from fine-tuning. The overall performance of all 12 fine-tuned ProPicker-C pickers is similar to that of a single DeepFinder trained on all 12 classes.

# F    PICKING RIBOSOMES AND APOFERRITIN IN A DIVERSE REAL-WORLD TOMOGRAM

Both real-world experiments presented in Section 4.1.3 are on tomograms containing mostly ribosomes, which have a large molecular weight and are, therefore, comparably easy to pick. Here, we show that ProPicker is able to pick particles other than ribosomes in real-world tomograms based

on a single prompt as well. We also show that ProPicker is able to distinguish particles in a diverse real-world tomogram.

We consider a tomogram from the DS-10440 dataset generated by Peck et al. (2024), which we downloaded from the Cryo ET Data Portal (Ermel et al., 2024). Among other particles, the dataset contains expert annotations for cytosolic ribosomes and apoferritin. Apoferritin has a molecular weight of 450 kilodalton which is $10.47\%$ of that of the ribosomes in the dataset (4.3 megadalton), and is also smaller in size than the ribosomes (Peck et al., 2024). Our goal is to pick all instances of the ribosomes and the apoferritin in the same tomogram based on a single prompt.

As can be seen in Figure A4, the segmentation model of ProPicker is able to detect both the ribosomes (ice blue) and the apoferritin (pink) based on a single prompt each. Using ProPicker-TM, i.e. using TomoTwin to pick particles in the respective segmentaiton masks, we obtained a best case F1 picking score of 0.69 for the ribosomes and 0.71 for the apoferritin. The cluster-based picking approach of ProPicker-C gave a best-case F1 picking score of 0.5 and 0.53 respectively. The worse performance of cluster-based picking is likely due to crowded parts (see, e.g. the zoomed-in region), where several instances of the same particle appear close together, which makes clustering challenging compared to peak-finding, which is used in TomoTwin-based picking. TomoTwin on its own, i.e. applied to the entire tomogram, performs similarly to ProPicker-TM.

