# OpenReview forum: "ProPicker: Promptable Segmentation for Particle Picking in Cryogenic Electron Tomography"
_ICLR.cc/2025/Conference — ICLR 2025 Conference Withdrawn Submission_

### Official Review · Reviewer_crDQ · 2024-10-28

**Soundness:** 2
**Presentation:** 3
**Contribution:** 2
**Rating:** 3
**Confidence:** 4

**Summary:**

The paper presents a 3D U-Net based pipeline for particle picking, i.e. object detection, in cryo electron tomography based on a prompt in the form of a reference image of a sought particle. The pipeline employs an off-the-shelf CNN from the current state of the art in the field, TomoTwin, to embed the prompt image; It then uses the U-Net along with a conditioning mechanism in the decoder, FiLM, to generate a segmentation map of the prompted particle; The foreground of this map is then leveraged to reduce the search space of the sought particle, either by using each connected component as a candidate location, or by using the full foreground in a traditional template matching approach.  The pipeline achieves a significant speed-up over TomoTwin, which uses a sliding window CNN in place of a U-Net in an overall otherwise similar pipeline.

**Strengths:**

The work is clearly written and easy to follow. The speedup over TomoTwin is clearly demonstrated and of clear practical relevance.

**Weaknesses:**

The reported speedup, while practically relevant, appears methodologically fully due to replacing a sliding window CNN with a U-Net. This is methodologically not new, as it was the core contribution of the FCN [Long et al., Fully Convolutional Networks for Semantic Segmentation, CVPR 2015] and further refined by the original U-Net work as cited by the authors. Thus the work appears to be out-of-scope for ICLR due to the absence of methodological novelty, while a more application-focused publication venue could be very fitting.

Furthermore, the work appears to neglect previous work on formal properties of encoder-decoder architectures of the U-Net: The authors perform a "stride analysis" to find the best trade-off between window tiling (i.e., chopping the large input image into digestible pieces) and accuracy of the U-Net. (Outputs are then stitched back together, with averaging applied in overlapping regions of the output tiles.) The authors report that larger strides / less overlap in general lead to less accurate predictions. However, it has been shown that U-Nets can be run in a tile-and-stitch manner that is provably equivalent to processing full images as a whole [Rumberger et al., How Shift Equivariance Impacts Metric Learning for Instance Segmentation, ICCV 2021], thus enabling sound processing of arbitrarily large / too-large-for-GPU-memory images. To this end, the U-Net needs to be applied with valid padding and feasible output window size. This approach renders tile-and-stitch processing with a U-Net periodic-k shift equivariant, which may alleviate the need for any output tile overlap in case of the authors' application. Note, there is still a chance though that averaging over test time augmentations in the form of input image shifts by <k pixels may improve accuracy. Both should be tried out by the authors for a formally sound analysis of the impact of window tiling on particle presence map accuracy and speed.

**Questions:**

What kind of padding do you employ in the U-Net? What is the input- (and output-) window size?
Please try valid padding in combination with input window size as large as your GPU can take, and output window cropping to the closest multiple of 2^l, where l is the number of pooling layers in your U-Net (I believe 5?), as described in Rumberger et al., 2021.

---

> ### Author Response · Authors · 2024-11-26
> **Response**
>
> We thank the reviewer for the valuable feedback and questions. We are happy to hear the reviewer finds our focus on fast picking practically relevant. Here is our point-by-point response:
>
> ```
> The reported speedup, while practically relevant, appears methodologically fully due to replacing a sliding window CNN with a U-Net. This is methodologically not new, as it was the core contribution of the FCN [Long et al., Fully Convolutional Networks for Semantic Segmentation, CVPR 2015] and further refined by the original U-Net work as cited by the authors. Thus the work appears to be out-of-scope for ICLR due to the absence of methodological novelty, while a more application-focused publication venue could be very fitting.
> ```
>
> The use of FCNs and U-Nets for fast segmentation has indeed been extensively explored in both the general and the cryo-ET literature on segmentation. However, aside from CryoSAM (which is significantly different from ProPicker, we refer to “General Response 1”), promptable segmentation has not been investigated. We regard our approach to achieve fast and universal picking by conditioning a U-Net on a general particle encoder (TomoTwin) for promptable segmentation as novel.
>
> ```
> Furthermore, the work appears to neglect previous work on formal properties of encoder-decoder architectures of the U-Net: The authors perform a "stride analysis" to find the best trade-off between window tiling (i.e., chopping the large input image into digestible pieces) and accuracy of the U-Net. (Outputs are then stitched back together, with averaging applied in overlapping regions of the output tiles.) The authors report that larger strides / less overlap in general lead to less accurate predictions. However, it has been shown that U-Nets can be run in a tile-and-stitch manner that is provably equivalent to processing full images as a whole [Rumberger et al., How Shift Equivariance Impacts Metric Learning for Instance Segmentation, ICCV 2021], thus enabling sound processing of arbitrarily large / too-large-for-GPU-memory images. To this end, the U-Net needs to be applied with valid padding and feasible output window size. This approach renders tile-and-stitch processing with a U-Net periodic-k shift equivariant, which may alleviate the need for any output tile overlap in case of the authors' application. Note, there is still a chance though that averaging over test time augmentations in the form of input image shifts by <k pixels may improve accuracy. Both should be tried out by the authors for a formally sound analysis of the impact of window tiling on particle presence map accuracy and speed.
> ```
>
> We thank the reviewer for pointing out this relevant paper by Rumberger et al. on an alternative to our approach of applying the U-Net in a sliding window manner.
>
> We found that our sliding window approach (which is also used by many other segmentation-based pickers) is a fast way to produce spatially consistent 3D segmentation masks that translate to very high picking performance. However, we appreciate the reviewer’s suggestion which can potentially speed-up ProPicker even further, and we have cited Rumberger et al. in the revised paper.
>
> **Response to the Reviewer's Questions:**
>
> ```
> What kind of padding do you employ in the U-Net? What is the input- (and output-) window size? Please try valid padding in combination with input window size as large as your GPU can take, and output window cropping to the closest multiple of 2^l, where l is the number of pooling layers in your U-Net (I believe 5?), as described in Rumberger et al., 2021.
> ```
>
> Our U-Net has 5 pooling layers, takes a cubic sub-tomogram of size 64x64x64 as input and returns a cube of the same shape. We did not employ any padding.
> For our comment on the proposed experiment, we refer to our answer above.

---

> > ### Comment · Reviewer_crDQ · 2024-12-03
> >
> > Many thanks for your point-by-point response.
> >
> > My concerns about novelty and formal soundness remain and I am thus keeping by score.
> >
> > In particular, performing an empirical stride analysis for tile-and-stitch inference is not the formally sound way to go -- instead, the U-Net first needs to be run the "correct" way, i.e., tile-and-stitch inference with appropriate output window cropping, thus equivalent to processing arbitrarily large input images in one go; Following up on this baseline, a stride analysis can still be performed, but this is then a clean, separate analysis to measure the effect of Test Time Augmentation (=offset transformations of input images).
> >
> > For completeness, here's another reference that you might find helpful:
> >
> > Possolo M, Bajcsy P. Exact Tile-Based Segmentation Inference for Images Larger than GPU Memory. J Res Natl Inst Stand Technol. 2021 Jun 3;126:126009. doi: 10.6028/jres.126.009. PMID: 39015626; PMCID: PMC10914126.
> >
> > It describes correct tile-and-stitch as in Rumberger 2021 (albeit without proof of equivalence to processing full images in one go)

---

### Official Review · Reviewer_2VwP · 2024-11-02

**Soundness:** 3
**Presentation:** 3
**Contribution:** 1
**Rating:** 5
**Confidence:** 3

**Summary:**

This paper proposes a new promptable method for particle picking in cryo-ET data. By using the off-the-shelf encoder TomoTwin, it encodes a manually selected prompt into an embedding, which is then injected into the decoder of a 3D U-Net to achieve prompt-based particle segmentation. The authors demonstrate the method's advantages in speed and discuss its generalization capabilities and applicability to real experimental data.

**Strengths:**

- Through an additional U-Net, this method enables TomoTwin to perform particle picking with a larger stride, thereby achieving a speed increase while maintaining on-par accuracy.
- The paper discusses the method's generalization performance and demonstrates its effectiveness on real experimental data, indicating its practical value.

**Weaknesses:**

- The paper claims promptable, fast and universal particle picking, but there is already closely related work in this area that is neither cited nor compared in this study.
[1] CryoSAM: Training-Free CryoET Tomogram Segmentation with Foundation Models. MICCAI 2024
[2] Accurate Detection of Proteins in Cryo-Electron Tomograms from Sparse Labels. ECCV 2022
- This paper relies on a robust prompt encoder, TomoTwin, which itself is already capable of generalized reference-based or clustering-based particle picking. A key question is whether the model's generalization capability mainly stems from this strong prompt encoder. The addition of the extra U-Net could potentially undermine TomoTwin’s inherent generalization ability by introducing layers that may reduce the flexibility of the learned embeddings, potentially limiting adaptability to unseen particles.
- The method shows some generalization capability on unseen particles but is quite sensitive to variations among different particles, with performance dropping significantly on certain particles. Fine-tuning can mitigate this issue; however, the need for fine-tuning contradicts the claim of universal picking.
- Figure 3 also reveals considerable sensitivity to prompt choice for many proteins, with several outliers showing low performance. The reasons behind these low-performing outliers need further exploration, including a more detailed description of how prompt choices are selected and what is a good choice supposed to be like. The authors recommend trying a few prompts for each particle, if prompt selection relies solely on random attempts, the practical usability of the method is compromised.

**Questions:**

- Why does each chart in the experimental results display different proteins? Is it possible to unify the experiments and compare them on the same proteins?
- In Section 4.6, the visualization results of the experimental data are quite difficult to compare. For example, in Figure 6, from my perspective, ProPicker has missed recalling quite a few objects that appear to be white particles. Additionally, this section compares the IoU of TomoTwin, ProPicker, with the expert labels, but IoU may not be a suitable metric for this task. On one hand, it only reflects the overlap of the global mask and does not indicate the proportion of objects that have been picked; on the other hand, the precision of the expert labeled mask does not seem very high from the figure, making it inappropriate to pursue a high IoU blindly. Metrics similar to average precision at certain IoU threshold in object detection would better reflect the issues at hand.

---

> ### Author Response · Authors · 2024-11-26
> **Response (Part 1)**
>
> We thank the reviewer for the valuable feedback and questions. Here is our point-by-point response:
>
> ```
> The paper claims promptable, fast and universal particle picking, but there is already closely related work in this area that is neither cited nor compared in this study. [1] CryoSAM: Training-Free CryoET Tomogram Segmentation with Foundation Models. MICCAI 2024 [2] Accurate Detection of Proteins in Cryo-Electron Tomograms from Sparse Labels. ECCV 2022
> ```
> Regarding CryoSAM, we refer the reviewer to “Shared Response 1”.
> We thank the reviewer for pointing out the second paper by Huang et al.The paper is not on universal or promptable picking but discusses how to train a particle picker on little data. Although is not the focus of our paper (we refer to “Shared Response 2”), the paper is relevant to our work and we cite it in the revised paper.
>
> ```
> This paper relies on a robust prompt encoder, TomoTwin, which itself is already capable of generalized reference-based or clustering-based particle picking. A key question is whether the model's generalization capability mainly stems from this strong prompt encoder. The addition of the extra U-Net could potentially undermine TomoTwin’s inherent generalization ability by introducing layers that may reduce the flexibility of the learned embeddings, potentially limiting adaptability to unseen particles.
> ```
>
> As, in its current form, ProPicker has been trained to work with TomoTwin as prompt encoder, the performance of ProPicker is indeed correlated with that of TomoTwin. We observe similar performance and robustness for almost all particles in both the synthetic and the real world tomograms.
> ProPicker can be seen as a framework to significantly speed up template matching-based picking (in this case with TomoTwin), while maintaining its performance.
> ```
> The method shows some generalization capability on unseen particles but is quite sensitive to variations among different particles, with performance dropping significantly on certain particles. Fine-tuning can mitigate this issue; however, the need for fine-tuning contradicts the claim of universal picking.
> ```
>
> Generalization and robustness is a challenge for all deep learning based pickers, and performance drops are not unique to ProPicker. We refer the reviewer to the discussion by Huang et al. (Huang, Qinwen, Ye Zhou, and Alberto Bartesaghi. "MiLoPYP: self-supervised molecular pattern mining and particle localization in situ." Nature Methods (2024): 1-10.). Moreover we added a new result showing that TomoTwin suffers from similar performance drops on unseen particles which have been excluded from its training set (see Figure 5 in the revised paper).
>
> It is commonly accepted in the machine learning community that robustness and generalization can be significantly improved by training on larger and more diverse datasets. As discussed in the paper, such datasets are currently being collected and we expect both TomoTwin and ProPicker to become significantly more robust when training on more real world data. Fine-tuning is always likely to improve performance but we expect the impact of fine-tuning to diminish as more training data becomes available.
>
> ```
> Figure 3 also reveals considerable sensitivity to prompt choice for many proteins, with several outliers showing low performance.
> The reasons behind these low-performing outliers need further exploration, including a more detailed description of how prompt choices are selected and what is a good choice supposed to be like.
> The authors recommend trying a few prompts for each particle, if prompt selection relies solely on random attempts, the practical usability of the method is compromised.
> ```
>
> We agree that the choice of the prompt is an important “hyperparameter” of our method and we appreciate the reviewer’s observation.
>
> We provide some additional context regarding Figure 3 (Figure A.1 in the Appendix in the revised paper): The figure shows picking results for a total of 200 prompts (20 particles and 10 prompts per particle). Out of these 200 prompts, only 14 (7%) are classified as lower-end outliers by the “1.5 times inner-quartile-range” rule. Most of these outliers are still close to the mean.
>
> Slight variations of the performance are due to different orientations of the particles and randomness of the noise. The rare larger drops are likely caused by additional content inside the 37x37x37 box that contains the prompt particle. Such additional content can include parts of other particles, parts of vesicles, or parts of filaments, and has an adversarial effect on the prompt encoder and therefore also ProPicker. A user can easily avoid such bad prompts by inspection.

---

> ### Author Response · Authors · 2024-11-26
> **Response (Part 2)**
>
> **Response to the Reviewer's Questions:**
>
> ```
> Why does each chart in the experimental results display different proteins? Is it possible to unify the experiments and compare them on the same proteins?
> ```
>
> We agree that this can be a bit confusing. Unfortunately, using the same particles for all experiments, is not possible, because each experiment highlights a specific aspect of ProPicker which is often related to the specific particles. E.g. for the fine-tuning experiments (Figure 5) we need particles where ProPicker’s out of the box performance is low compared to the good performance on the particles in Table 1.
>
> ```
> In Section 4.6, the visualization results of the experimental data are quite difficult to compare. For example, in Figure 6, from my perspective, ProPicker has missed recalling quite a few objects that appear to be white particles. Additionally, this section compares the IoU of TomoTwin, ProPicker, with the expert labels, but IoU may not be a suitable metric for this task. On one hand, it only reflects the overlap of the Shared mask and does not indicate the proportion of objects that have been picked; on the other hand, the precision of the expert labeled mask does not seem very high from the figure, making it inappropriate to pursue a high IoU blindly. Metrics similar to average precision at certain IoU threshold in object detection would better reflect the issues at hand.
> ```
>
> We agree with the reviewer and have therefore replaced the IoU-based comparison with a comparison of the F1 score with respect to the expert annotations. Both ProPicker and TomoTwin achieve 0.6.
> We agree that the expert annotations seem incomplete. However, De Teresa-Trueba et al. who proposed the dataset also report the F1 with respect to the expert annotations for their DeePiCt method, which is a picker similar to DeepFinder that has been specifically trained on EMPIAR 10988. For more details, we refer to the DeePiCt paper [1].
>
> [1] De Teresa-Trueba, Irene, et al. "Convolutional networks for supervised mining of molecular patterns within cellular context." Nature Methods 20.2 (2023): 284-294.

---

> > ### Comment · Reviewer_2VwP · 2024-11-29
> >
> > Thank you for your detailed point-by-point response. My primary concern lies with the paper's claim of proposing a "promptable, fast, and universal picker". The design of a promptable system typically assumes a model with strong few-shot or even zero-shot capabilities, demonstrating significant generalization ability (universal). If ProPicker merely accelerates the TomoTwin method without contributing to the generalization aspect, then the contribution of this paper is limited. As we all know, fine-tuning can enhance a model's performance on unseen data, but this doesn't actually provide a contribution towards universal picking. Given that, and considering the standards for ICLR submissions, I feel that its contribution does not sufficiently meet the acceptance bar. I will maintain my score.

---

### Official Review · Reviewer_Rmoy · 2024-11-03

**Soundness:** 1
**Presentation:** 1
**Contribution:** 1
**Rating:** 3
**Confidence:** 5

**Summary:**

The paper proposes a cryo-ET particle-picking method called ProPicker, which is a promptable encoder-decoder-based segmentation model. ProPicker is trained on synthetic data and can be finetuned on experimental data with few prompts. The prompts are processed with a prompt encoder which is further used as input to the decoder of the segmentation model. With such a strategy, ProPicker demonstrated fast prompt-based picking on several synthetic and experimental cryo-ET datasets.

**Strengths:**

The authors proposed a working learning-based solution for particle picking in cryo-ET, which is an important problem in cellular and structural biology.

**Weaknesses:**

*Novelty: I highly doubt whether there is any actual novelty in this work. Prompt-based segmentation for particle picking has been already introduced for cryo-ET (See CryoSAM, MICCAI 2024).

*No discussion and comparison with related works: There are several learning-based particle-picking methods for cryo-ET now. The mentionable ones are DeepETPicker, CryoSAM, Huang et. al. (ECCV, 2022), CrYOLO, etc. However, the authors did not mention them at all, let alone provide any comparison with them. They only used TomoTwin as baseline.

*Evaluation: The quantitative evaluation of real data is questionable. Since there are annotations on the tomograms,  the authors could have simply calculated the F1 score as they did on synthetic data. Instead, the authors used the IoU score for the evaluation of real data, which does not make much sense, since the annotations are points, not volumes. Moreover, the IoU score is very small. The authors claimed that expert annotations miss some particles without properly validating such claims. The visualization in Figure 7 is simply not enough as a blob-like shape may or may not be a particle of interest. Moreover, there are also FAS particles apart from ribosomes in EMPIAR-10988. One way to validate the author’s claim is to perform subtomogram averaging (STA). The authors talked about STA in the appendix but did not perform it for their predictions.

*Depends on Denoising: While finetuned on real tomograms, the authors performed denoising. For EMPIAR 10045, the tomograms were denoised with CryoCARE. For EMPIAR 10988, the tomograms (did not mention if only VPP, only defocus, or both) are denoised with Topaz. ProPicker was applied to the denoised tomograms. Such dependence on denoising methods is a significant limitation for ProPicker. Moreover, different denoising methods were used for different tomograms- which method to use can be a difficult choice.

*Usability: The authors argued that many learning-based picking methods require particle radius as input which limits their usability. However, the authors initiated another parameter stride value (s), which is crucial for ProPicker to work. The authors used very different s value for baseline TomoTwin and their method ProPicker. From the choice of values they made for ProPicker, it seems like they are just using the approximate diameter of the ribosome as pixels as the stride value. In other words, they are indirectly using the radius, which according to their claims will limit the usability of the method.

**Questions:**

* How do you know the optimal stride (s) value? You used a very small (s) value for TomoTwin but a large stride value for ProPicker, why?
Also, the stride value used for ProPicker is similar to the diameter of the ribosome in pixels. In such a case, you indirectly need an idea of the particle radius to select the optimal stride value. Is this the case?
* EMPIAR 10988 also contains FAS. I wonder why ProPicker was not tested for FAS picking. Can you check your picking performance for FAS particles?
* You mentioned authors missed many True particles for EMPIAR 10988 that ProPicker picked. However, such a statement is not experimentally validated. One way is to perform subtomogram averaging (with the same parameters) for both author-annotated and ProPicker-annotated particles and see if ProPicker improves resolution.
* How did you finetune on real tomograms from EMPIAR 10045 and EMPIAR 10988? How many tomograms did you use? How many prompts did you use? For EMPIAR 10988, did you use VPP tomograms or Defocus-only tomograms?
* How dependent is ProPicker’s performance on denoising?
* You mentioned prompts can be amino acid sequences or atomic coordinates as well. How does that work?

---

> ### Author Response · Authors · 2024-11-26
> **Response (Part 1)**
>
> We thank the reviewer for the valuable feedback and questions. Here is our point-by-point response:
>
> ```
> *Novelty: I highly doubt whether there is any actual novelty in this work. Prompt-based segmentation for particle picking has been already introduced for cryo-ET (See CryoSAM, MICCAI 2024).
> ```
> We refer the reviewer to “Shared Response 1”.
>
> ```
> *No discussion and comparison with related works: There are several learning-based particle-picking methods for cryo-ET now. The mentionable ones are DeepETPicker, CryoSAM, Huang et. al. (ECCV, 2022), CrYOLO, etc. However, the authors did not mention them at all, let alone provide any comparison with them. They only used TomoTwin as baseline.
> ```
>
>
> We refer the reviewer to “Shared Response 2”. We emphasize that we do not only compare to TomoTwin but also to DeepFinder (which is conceptually very close to the baselines the reviewer has pointed out). We have extended this comparison further in the revised paper (see Section 4.2).
>
> ```
> *Evaluation: The quantitative evaluation of real data is questionable. Since there are annotations on the tomograms, the authors could have simply calculated the F1 score as they did on synthetic data. Instead, the authors used the IoU score for the evaluation of real data, which does not make much sense, since the annotations are points, not volumes. Moreover, the IoU score is very small. The authors claimed that expert annotations miss some particles without properly validating such claims. The visualization in Figure 7 is simply not enough as a blob-like shape may or may not be a particle of interest. Moreover, there are also FAS particles apart from ribosomes in EMPIAR-10988. One way to validate the author’s claim is to perform subtomogram averaging (STA). The authors talked about STA in the appendix but did not perform it for their predictions.
> ```
>
> Our IoU-based comparison is sensible because EMPIAR-10988 contains segmentation masks that have been created by experts. The IoU scores have been calculated with respect to these masks rather than points.
>
> To make the evaluation of the real-world experiments more consistent with the remainder of the paper, we have now replaced the evaluation based on IoU with F1 picking scores for EMPIAR-10988. Both TomoTwin and ProPicker achieve 0.6.
>
> We also verified that none of the particles picked by ProPicker is FAS.
>
> ```
> *Depends on Denoising: While finetuned on real tomograms, the authors performed denoising. For EMPIAR 10045, the tomograms were denoised with CryoCARE. For EMPIAR 10988, the tomograms (did not mention if only VPP, only defocus, or both) are denoised with Topaz. ProPicker was applied to the denoised tomograms. Such dependence on denoising methods is a significant limitation for ProPicker. Moreover, different denoising methods were used for different tomograms- which method to use can be a difficult choice.
> ```
>
> The results on the two real-world datasets have been achieved by performing inference with ProPicker with a single prompt without any fine-tuning.
>
> As stated in the paper, we denoised the real-world tomograms using a simple and efficient Gaussian filter. We did apply CryoCARE to EMPIAR 10045 and Topaz to EMPIAR 10988 only to have cleaner tomograms for our figures.
>
> ```
> *Usability: The authors argued that many learning-based picking methods require particle radius as input which limits their usability.
> ```
> We do not claim that requiring knowledge of the particle size is a limitation for many pickers, and in fact ProPicker assumes the particle size to be known in our problem statement (Section 3.1).
> ```
> However, the authors initiated another parameter stride value (s), which is crucial for ProPicker to work. The authors used very different s value for baseline TomoTwin and their method ProPicker.
> ```
> As discussed and shown in Section 4.1.1 of the revised paper (Section 4.1 in the original paper), ProPicker’s performance is not particularly sensitive to the stride.
>
> We show that TomoTwin needs small strides to achieve good performance. This is due to it being a template matching-like method. The fact that ProPicker can achieve good performance with large strides is a key advantage of our method over template-matching methods. All of this is discussed in detail in the paper (Section 2, Section 4.1.1).
>
> ```
> From the choice of values they made for ProPicker, it seems like they are just using the approximate diameter of the ribosome as pixels as the stride value. In other words, they are indirectly using the radius, which according to their claims will limit the usability of the method.
> ```
>
> As can be seen in Figure 2, ProPicker achieves good results on 100 unique particles with vastly different sizes when using stride 32 or even 56. This clearly demonstrates that aligning particle size and stride length is not necessary for ProPicker.
> We have not deliberately chosen ProPicker’s stride according to the particle size in any of our experiments.

---

> ### Author Response · Authors · 2024-11-26
> **Response (Part 2)**
>
> **Response to the Reviewer's Questions:**
>
> ```
> How do you know the optimal stride (s) value? You used a very small (s) value for TomoTwin but a large stride value for ProPicker, why? Also, the stride value used for ProPicker is similar to the diameter of the ribosome in pixels. In such a case, you indirectly need an idea of the particle radius to select the optimal stride value. Is this the case?
> ```
> We refer to the reviewer to our comment on the reviewer’s concern regarding the usability of ProPicker.
>
> ```
> EMPIAR 10988 also contains FAS. I wonder why ProPicker was not tested for FAS picking. Can you check your picking performance for FAS particles?
> ```
>
> We found that neither ProPicker nor TomoTwin is able to pick FAS in EMPIAR 10988. However, FAS is known to be very hard to pick even for pickers that have been explicitly trained on EMPIAR 10988 to specifically pick FAS, as discussed in the DeePiCT paper [1].
>
> [1] De Teresa-Trueba, Irene, et al. "Convolutional networks for supervised mining of molecular patterns within cellular context." Nature Methods 20.2 (2023): 284-294.
>
>
> ```
> You mentioned authors missed many True particles for EMPIAR 10988 that ProPicker picked. However, such a statement is not experimentally validated. One way is to perform subtomogram averaging (with the same parameters) for both author-annotated and ProPicker-annotated particles and see if ProPicker improves resolution.
> ```
>
> Although Reviewer 2VwP raised a similar concern: “the precision of the expert labeled mask does not seem very high from the figure”, we have decided to remove the statement that the expert masks seem incomplete. Instead, we follow De Teresa-Trueba et al. who proposed the dataset in the DeePiCt paper [1], and report the picking F1 score with respect to the expert annotations. Both ProPicker and TomoTwin achieve 0.6.
>
> [1] De Teresa-Trueba, Irene, et al. "Convolutional networks for supervised mining of molecular patterns within cellular context." Nature Methods 20.2 (2023): 284-294
>
>
> ```
> How did you finetune on real tomograms from EMPIAR 10045 and EMPIAR 10988? How many tomograms did you use? How many prompts did you use? For EMPIAR 10988, did you use VPP tomograms or Defocus-only tomograms?
> ```
>
> The results on the two real-world datasets have been achieved by performing inference with ProPicker with a single prompt without any fine-tuning. For EMPIAR 10988 we used the defocus-only tomograms.
>
> ```
> How dependent is ProPicker’s performance on denoising?
> ```
> For all evaluation on synthetic data, we did not use any denoising. On EMPIAR 10988, ProPicker achieves a picking F1 score of 0.55 w.r.t. the expert annotations when applied to the un-denoised tomogram vs. 0.61 when we denoise the tomogram by convolution with a Gaussian kernel (as in the paper).
>
> ```
> You mentioned prompts can be amino acid sequences or atomic coordinates as well. How does that work?
> ```
> This could be achieved by replacing the TomoTwin prompt encoder with another encoder that maps either amino acid sequences or coordinates into a 1D feature space and re-training ProPicker based on these prompts. We leave this for future work.

---

> ### Comment · Reviewer_Rmoy · 2024-11-28
>
> Thanks to the authors for the rebuttal and point-by-point response. The authors claried why novelty concerns should be minimized. However, concerns about the evaluation on real-world tomograms remain unresolved. The authors avoided comparison with DeepETPicker and CrYOLO, citing that these are not universal pickers, unlike their method. Yet, if ProPicker is truly universal, it raises questions about its inability to pick FAS in S. pombe, where yeast FAS is approximately 1 MDa. This suggests that the "universal picking" capability may be limited to simulated datasets, which are inherently less complex and do not fully represent real tomograms' challenges.
>
> Additionally, the F1 score for ribosome prediction in EMPIAR-10988 lacks clarity. While the response mentions an F1 score of 0.61, the revised manuscript states it as the "best-case" F1 score. The definition of "best-case" remains ambiguous. Another concern is the use of only defocus tomograms, despite VPP tomograms being better suited for particle picking. The dataset contains 10 VPP tomograms and 10 defocus tomograms, but it is unclear whether results are reported for a single tomogram or averaged across all 10.
> Finally, subtomogram averaging results are notably absent. For practical applications in biology, methods should prioritize yielding particles that result in higher-resolution subtomogram averages, rather than focusing solely on fast inference that compromises final resolution. Given these concerns, the initial rating remains unchanged, as the evaluation and application perspectives still require significant improvement.

---

### Official Review · Reviewer_wWtW · 2024-11-04

**Soundness:** 2
**Presentation:** 3
**Contribution:** 1
**Rating:** 3
**Confidence:** 4

**Summary:**

This paper introduced a particle picking method for cryo-electron tomography (cryo-ET), ProPicker, that formulate particle picking in cryo-ET as a promptable segmentation problem. Particle picking in cryo-ET is a challenging problem and is also one of the bottlenecks in the cryo-ET data processing. The authors claims that ProPicker is a universal model that balances the speed and performance, by being significantly faster than other methods, while having similar picking accuracies.

**Strengths:**

ProPicker as a particle picking application seems to be significantly faster than other methods, which is an advantage for the end users. The promptable design inspired by SAM also provides good flexibility so that users can use it for different targets to explore their tomograms.

**Weaknesses:**

1. The main weakness of this manuscript as a submission of ICLR is the lack of novelty in the methods. It seems that the major difference between ProPicker and other deep learning-based picking methods are the promptable design. The encoder for the prompt is taken from TomoTwin as-is, the segmentation model is a standard 3D U-Net, and the particle coordinate finding uses clustering or template matching which both are very standard procedures in other similar methods. The SAM-inspired promptable design itself is not completely novel in the cryo-EM/ET field either, as there are similar methods like cryoSAM [1, 2].

2. Not enough performance comparison with other methods. The authors mainly used TomoTwin as the baseline comparison, where there are other modern methods like DeepETPicker, DeePiCt, cryoSAM. Also I noticed that the authors used TomoTwin as the baseline for some tasks, while switched to DeepFinder for other tasks, which is not satisfying. How does ProPicker compare with them in terms of performance and speed on both synthetic data and real data?

3. In many cases, the ultimate goal of cryo-ET data processing is to achieve the best reconstruction. Using the picked particles by ProPicker or by other methods in real data, how do the reconstructions look like? Will ProPicker's reconstruction achieves a better or on-par resolution?

4. The authors presented two real data cases, but in both of them the targets are ribosomes. Ribosome is well-known as an easy target for picking due to their relatively large size and being abundance in the in situ data. The authors should present at least one other real case in which the picking target is not ribosome.

5. For the related works, there should also be some brief mentions of particle picking methods for single-particle cryo-EM, as these two problems are very closely related.

6. There are no available ablation studies in the manuscript; for example. I feel that the discussion about the stride s should be reformatted into an ablation study section for a better reading. If there are other hyperparameters, it should be discussed as well.

[1] Zhao, Yizhou, et al. "CryoSAM: Training-Free CryoET Tomogram Segmentation with Foundation Models." International Conference on Medical Image Computing and Computer-Assisted Intervention. Cham: Springer Nature Switzerland, 2024.
[2] He, Fei, et al. "Adapting Segment Anything Model (SAM) through Prompt-based Learning for Enhanced Protein Identification in Cryo-EM Micrographs." arXiv preprint arXiv:2311.16140 (2023).

**Questions:**

How does the missing wedge artifact affect ProPicker? To elaborate, the tomogram will suffer from the missing wedge artifact, and so will be the prompt if it is a subtomogram cropped from the larger tomogram.

---

> ### Author Response · Authors · 2024-11-26
> **Response (Part 1)**
>
> We thank the reviewer for the valuable feedback and questions. Here is our point-by-point response:
>
> ```
> 1. The main weakness of this manuscript as a submission of ICLR is the lack of novelty in the methods. It seems that the major difference between ProPicker and other deep learning-based picking methods are the promptable design. The encoder for the prompt is taken from TomoTwin as-is, the segmentation model is a standard 3D U-Net, and the particle coordinate finding uses clustering or template matching which both are very standard procedures in other similar methods. The SAM-inspired promptable design itself is not completely novel in the cryo-EM/ET field either, as there are similar methods like cryoSAM [1, 2].
> ```
>
> We agree that the main novelty of ProPicker is the promptable design and our concrete combination of the flexible TomoTwin method with a fast 3D segmentation model. While the general idea of using promptable segmentation has been explored by Zhao et al. in the CryoSAM paper, ProPicker’s design is significantly different from CryoSAM and enables better picking in diverse environments (see Shared Response 1).
>
> The second SAM-based method mentioned by the reviewer (Fei et al.) is for picking particles in 2D single-particle cryo-EM micrographs. Compared to ProPicker, the paper considers a different domain (2D micrographs vs. 3D tomograms) and does not target universal picking based on prompting, but applies prompt learning to SAM,  which is an alternative to fine-tuning.
>
> ```
> 2. Not enough performance comparison with other methods. The authors mainly used TomoTwin as the baseline comparison, where there are other modern methods like DeepETPicker, DeePiCt, cryoSAM. Also I noticed that the authors used TomoTwin as the baseline for some tasks, while switched to DeepFinder for other tasks, which is not satisfying. How does ProPicker compare with them in terms of performance and speed on both synthetic data and real data?
> ```
> We refer the reviewer to “Shared Response 2”.
>
> ```
> 3. In many cases, the ultimate goal of cryo-ET data processing is to achieve the best reconstruction. Using the picked particles by ProPicker or by other methods in real data, how do the reconstructions look like? Will ProPicker's reconstruction achieves a better or on-par resolution?
> ```
>
> We agree that reconstruction experiments based on particles picked by ProPicker or other methods would be interesting.
>
> However, beyond particle reconstruction (via subtomogram averaging), there are other applications for particle picking, such as direct in-context interpretation of located particles, or analysis of the concentration or distribution of a particle type across the tomogram.
>
> Therefore, the direct evaluation of picking methods with metrics such as the F1 score, rather than reporting how they affect downstream tasks such as subtomogram averaging, is common in papers on particle picking(especially in the ML community), see for example the CryoSAM paper, or [1, 2].
>
>
> [1] Huang, Qinwen, et al. "Accurate detection of proteins in cryo-electron tomograms from sparse labels." European Conference on Computer Vision. Cham: Springer Nature Switzerland, 2022.
>
> [2] Uddin, Mostofa Rafid, et al. "TomoPicker: Annotation-Efficient Particle Picking in cryo-electron Tomograms." bioRxiv (2024): 2024-11.
>
> ```
> The authors presented two real data cases, but in both of them the targets are ribosomes. Ribosome is well-known as an easy target for picking due to their relatively large size and being abundance in the in situ data. The authors should present at least one other real case in which the picking target is not ribosome.
> ```
>
> We agree with the reviewer that testing on particles other than ribosomes would be interesting. Unfortunately, tomograms with ground truth annotated particles are extremely scarce and almost all of the annotations are ribosomes.
>
> We are currently working on a case study in which we apply ProPicker to a very recent real-world dataset that is part of the ongoing CryoET Data Portal challenge (https://cryoetdataportal.czscience.com/datasets/10440). The dataset contains a mixture of particles of varying size and molecular weight. Our initial results suggest that, out of this mixture, ProPicker is able to accurately pick both ribosomes and apoferritin based on a single prompt, without any fine-tuning. Apoferritin is significantly smaller and lighter (molecular weight 450 Kd) than the ribosomes we considered so far.
>
> **Edit (Nov. 28th):** We have added a new Appendix (Appendix F) in which we show the results from the case study described above.
>
> ```
> 3. For the related works, there should also be some brief mentions of particle picking methods for single-particle cryo-EM, as these two problems are very closely related.
> ```
>
> We have added a sentence to the beginning of the related work section (Section 2) in which we mention some popular deep learning-based pickers for single-particle cryo-EM.

---

> ### Author Response · Authors · 2024-11-26
> **Response (Part 2)**
>
> ```
> 6. There are no available ablation studies in the manuscript; for example. I feel that the discussion about the stride s should be reformatted into an ablation study section for a better reading. If there are other hyperparameters, it should be discussed as well.
> ```
>
> We discuss the stride s and its impact between performance and speed in detail in the main paper because the larger stride is what causes ProPicker to be faster than TomoTwin.
> Other hyperparameters for picking are three thresholds: One for the binarization of ProPicker’s sigmoid output, and two for the minimum and maximum cluster size to eliminate false positive clusters. Such hyperparameters are also used in other segmentation-based pickers like DeePiCt [2] or DeepETPicker [3] and have to be chosen for each particle of interest. In practice, the binarization threshold is commonly chosen by inspection and the thresholds for the cluster sizes are based on the size of the particle [1,2,3].
>
> [1] Rice, Gavin, et al. "TomoTwin: generalized 3D localization of macromolecules in cryo-electron tomograms with structural data mining." Nature methods 20.6 (2023): 871-880.
> [2] de Teresa-Trueba, Irene, et al. "Convolutional networks for supervised mining of molecular patterns within cellular context." Nature Methods 20.2 (2023): 284-294.
> [3] Liu, Guole, et al. "DeepETPicker: Fast and accurate 3D particle picking for cryo-electron tomography using weakly supervised deep learning." Nature Communications 15.1 (2024): 2090.
>
>
> --------
>
>
> **Response to Reviewers' Questions:**
> ```
> How does the missing wedge artifact affect ProPicker? To elaborate, the tomogram will suffer from the missing wedge artifact, and so will be the prompt if it is a subtomogram cropped from the larger tomogram.
> ```
>
> Both ProPicker and TomoTwin’s convolutional encoder which we use to encode the prompts
> have been trained and evaluated on tomograms with a missing wedge. We did not perform experiments on missing-wedge-corrected tomograms.

---

> > ### Comment · Reviewer_wWtW · 2024-12-02
> >
> > I would like to thank the authors for the response, yet I still have concerns about the novelty and the performance as a universal, promptable picker for cryo-ET. I am interested to see the performance of ProPicker on the CZI imaging challenge. I noticed the addition of DS-10440, however, the original dataset contains more protein species than ribosome and apoferritin. While I am fully aware that a universal picker for tomograms with great performance is difficult to achieve, I agree with the other reviewers on the weakness of this paper and decide to keep my score.

---

### Author Response · Authors · 2024-11-26
**Shared Response**

We thank all reviewers for their detailed feedback and their questions!

We respond to each reviewer individually, but we have distilled two main critiques shared by several reviewers, which we address in this shared response.

**Shared Concern 1:** Reviewers **wWtW, Rmoy** and **2VwP** argue that the promptable segmentation approach is not novel for cryo-ET particle picking because it has been proposed by Zhao et al. in CryoSAM.

**Shared Response 1:** We thank the reviewers for pointing out the CryoSAM paper, which we were not aware of and which, according to the ICLR review guidelines, is contemporaneous with our work (see last paragraph of this response). Now that we know about CryoSAM we, of course, carefully reviewed and tested it. We found that our ProPicker provides significant novelty compared to CryoSAM, and performs better.

As pointed out by the reviewers CryoSAM and ProPicker share the same general idea of using promptable segmentation for particle picking. **However, the crucial details of how promptable segmentation is achieved are vastly different and our approach is novel compared to CryoSAM:**
ProPicker has been specifically designed for the cryo-ET domain. It uses sub-tomograms as prompts and picks particles based on a 3D network that has been trained on domain-specific data, i.e., simulated cryo-ET tomograms. In contrast, CryoSAM relies on a general segmentation model (SAM) that has been trained to segment 2D natural images based on point prompts. Moreover, the DINOv2 features which CryoSAM uses for feature matching have again been learned from a large collection of (mostly “natural”) 2D images rather than being specifically tailored for cryo-ET.

When testing on synthetically generated clean ground-truth tomograms with 8 unique proteins (“clean” to provide a best-case scenario for CryoSAM which extensively denoised data), we found that CryoSAM has problems distinguishing the proteins from each other, while ProPicker, which has been trained on crowded data, achieves high picking F1 scores for all proteins even on noisy data.

**All discussions and comparisons related to CryoSAM can be found in Section 2, Section 4.1.2 and Appendix D in the revised paper.**

Finally, we would like to point out the following paragraph taken from the official ICLR 2025 reviewer guide (https://iclr.cc/Conferences/2025/ReviewerGuide):

```
Q: Are authors expected to cite and compare with very recent work? What about non peer-reviewed (e.g., ArXiv) papers? (updated on 7 November 2022)

A: We consider papers contemporaneous if they are published within the last four months. That means, since our full paper deadline is October 1, if a paper was published (i.e., at a peer-reviewed venue) on or after July 1, 2024, authors are not required to compare their own work to that paper. Authors are encouraged to cite and discuss all relevant papers, but they may be excused for not knowing about papers not published in peer-reviewed conference proceedings or journals, which includes papers exclusively available on arXiv. Reviewers are encouraged to use their own good judgement and, if in doubt, discuss with their area chair.

```

The CryoSAM has been published at this year's MICCAI which was held from October 6th to October 10th 2024 (CryoSAM was submitted to arXiv July 8th, 2024, https://arxiv.org/abs/2407.06833), making our work contemporaneous with CryoSAM. However, we still feel that the comparison has strengthened our work.

------

**Shared Concern 2:** Reviewers **wWtW** and **Rmoy** feel that our paper does not contain enough comparisons to existing methods and propose comparing ProPicker to methods like CrYOLO, DeepETPicker, and DeePiCt.

**Shared Response 2:**  We thank the reviewers for raising this concern, which indicates that we have not stated the scope of our work clearly and explicitly enough in the initial version of our paper: Our work targets fast and universal particle picking based on a single input prompt, without any training on the target tomograms. The state-of-the-art method for this is TomoTwin, and we compare to TomoTwin extensively.

CrYOLO, DeepETPicker, and DeePiCt are non-universal pickers and can only pick particles seen during training. Therefore, these baselines are only suitable for our experiments on fine-tuning ProPicker, as this is no longer universal picking. However, for fine-tuning, **we do compare to a state-of-the-art non-universal picker (DeepFinder) that is conceptually very similar to DeepETPicker and DeePiCt** (3D segmentation with a convolutional network + clustering or peak finding). **We have extended this comparison further in the revised version of our paper (see Section 4.2).**

Moreover, we have completely **re-structured the experiments section and for each experiment added a clear statement on which baselines we consider and why.**

---

### Note · Authors · 2024-12-10

I have read and agree with the venue's withdrawal policy on behalf of myself and my co-authors.